# Analysis of Thermal–Mechanical Properties of Silicon Dioxide/Polyvinylidene Fluoride Reinforced Non-Woven Fabric (Polypropylene) Composites

**DOI:** 10.3390/polym12020481

**Published:** 2020-02-21

**Authors:** Fangyun Kong, Mengzhou Chang, Zhenqing Wang

**Affiliations:** 1College of Aerospace and Civil Engineering, Harbin Engineering University, Harbin 150001, China; kongfangyun@hrbeu.edu.cn; 2College of Equipment Engineering, Shenyang Ligong University, Shenyang 110000, China; changmengzhou@hrbeu.edu.cn

**Keywords:** PP/SiO_2_/PVDF composite material, uniaxial tensile test, thermal–mechanical coupling, heat shrinkage property, microstructure characterization

## Abstract

In this paper, solution casting method is used to prepare the PP (polypropylene) non-woven fabric based composite film filled with silicon dioxide/polyvinylidene fluoride (SiO_2_/PVDF). The mechanical and thermodynamic properties of PP/SiO_2_/PVDF composites were studied by a uniaxial tensile test under different temperature and combustion experiment. It is found that the stress of PP/SiO_2_/PVDF composite film with 4 wt % SiO_2_ is the maximum value, reaching 18.314 MPa, 244.42% higher than that of pure PP non-woven. Meanwhile, the thermal–mechanical coupling tests indicate that with the increase of temperature, the ultimate stress and strain of the composite decrease. At the same time, the thermal shrinkage property of the composite during the heating process is studied. The modified composite has good thermal stability under 180 °C. Scanning electron microscope (SEM), X-ray diffraction (XRD) and thermogravimetric (TG) were used to characterize the pore shape, distribution and crystal phase change of the composite. The modified PP/SiO_2_/PVDF composite film structure shows high strength and good thermal stability, and can better meet the requirements of strength and thermal performance of lithium-ion battery during the charging and discharging process.

## 1. Introduction

Polyvinylidene fluoride (PVDF) is a copolymer of vinylidene fluoride (VDF) homopolymer and other small amount of fluorine-containing vinyl monomer. It is a high-quality fluoropolymer with both fluororesin and general-purpose resin characteristic. PVDF is one of the most widely used polymer materials because of its excellent comprehensive properties. 

First of all, the dielectric constant of PVDF film is relatively high. PVDF piezoelectric film is a new type of polymer sensitive material, which can be applied in actuators [1], sensors [2], memory [3] and simulation muscles. Zhu J et al. [4] prepared PVDF/TiO_2_ nanofiber piezoelectric material by electrospinning and assembled a sensor with sensor unit. The results show that the sensor has high sensitivity and will meet the requirements of high signal-to-noise ratio detector for deep exploration. Moreover, its low frequency cut off frequency is 3 Hz, which is better than that based on piezoelectric ceramics.

Secondly, PVDF is one of the main materials of fluorocarbon coatings [5], which can be used outdoors for a long time without aging and maintenance. Generally, it can be used in power stations, airports, highways, construction, car decoration and other fields.

Thirdly, the appearance of PVDF composite film provides a new way for medicine. PVDF material is flexible and can be used as the bone tissue cell [6]. Guo W et al. [7] developed a wearable cardiopulmonary signal sensor, which can monitor cardiopulmonary signal at any time in life.

Fourthly, the excellent mechanical properties of PVDF make it one of the new materials for strain observation and energy harvesting. Piezoelectric thin films have the ability to measure dynamic pressure by generating electrical response on the crystal surface, therefore, they can convert the external mechanical field into electrical signal. Dudem B et al. [8] have prepared PVDF piezoelectric materials, which can convert mechanical energy into electrical energy. The optimized piezoelectric materials are used to detect mechanical energy from vehicle motion and human motion effectively. Motta E P et al. [9] carried out cyclic loading test and uniaxial tensile test on PVDF flexible pipe. The results show that the elastic behavior does not depend on the strain rate, but the strain hardening caused by plastic deformation has rate dependence. Motta E P et al. proposed an elasto-viscoplastic mechanical model, which can describe the cyclic and monotonic inelastic behavior under the loading history.

Finally, PVDF based hydrophobic film is also used for oil/water separation, wastewater treatment, etc. Currently, the combination of PVDF ultrafiltration film and preactivated carbon filter was used to purify drinking water and it has become a new development direction [10]. Cui J et al. [11] prepared the PVDF@polydopamine(PDA)@SiO_2_ nano-composites, which have special surface micro nanostructures and pore induced capillarity. The nanocomposite membrane also showed excellent separation performance, regeneration ability and universality for oil–water emulsion separation. The membrane can be used for wastewater recovery and drinking water treatment. At the same time, PVDF film can also be used as pretreatment system of reverse osmosis in desalination. 

Composite materials are widely used in many fields, and many scholars and experts are engaged in the research of PVDF matrix composites. Li Y et al. [12] prepared PVDF hydrophobic flat composite membrane by non-solvent induced phase separation (NIPS) technology. The effects of three-dimensional Bi_2_WO_6_ on film morphology, hydrophobicity, thermal properties and permeability are studied. In 60 °C feed solution and 20 °C cold distilled water, the maximum permeation flux (about 13.12 kg/m^2^ h) of the membrane is higher than that of the pure PVDF membrane (12.06 kg/m^2^ h). In addition, the mixed membrane showed stability in the continuous desalination experiment. Damtie M M et al. [13] studied the influence of pH value and feed water composition on the membrane fouling thickness. It is found that the PVDF membrane with smooth surface had a thick fouling layer, which reduced the membrane flux. At higher pH value, PVDF membrane has lower organic water supply, which can obtain high quality permeate water. Therefore, this method can be used to determine the optimal membrane to meet the requirements of permeate flux and retention for the treatment of various organic pollutants in industrial wastewater.

High performance lithium ion batteries require a stable thermal working condition, electrochemical stability and good permeability of the separator. Shasha M et al. [14] have coated a composite layer composed of AlPO_4_, PVDF-hexafluoropropylene (HFP) and polymethyl methacrylate (PMMA) on the separation matrix of PP. The electrolyte absorption rate of coated PP is 2.28%. In addition, the separator has high peel strength and excellent thermal stability at 170 °C. When the coating separator is applied to LiNi_0.5_Co_0.2_Mn_0.3_O_2_ cathode battery, excellent cycle stability and rate performance can be observed. Jeong H S et al. [15] have prepared SiO_2_ reinforced heat-resistant polyethylene terephthalate (PET) non-woven/PVDF composite. The analysis of the microstructure indicates that the well connected gap between the added SiO_2_ nanoparticles and PVDF-HFP particles was formed. Compared with the commercial polyethylene (PE) separator, the material shows significant improvement in porosity, air permeability and electrolyte wettability, which is conducive to ion transport and delays the growth of battery impedance during the circulation process. Gong W et al. [16] have prepared a kind of poly (aryl ether sulfone ketone) (PPESK)/PVDF fiber membrane for lithium-ion battery by electrospinning technology. The prepared PPESK/PVDF fiber membrane has excellent thermal shutdown performance and thermal dimensional stability. The coin cell PVDF film shows excellent cycling performance, which indicates that the membrane may be used as the separator of high performance lithium-ion battery.

Therefore, in this paper, static mechanical properties and heat resistance of SiO_2_/PVDF composites have been studied to further develop other scholars’ research aimed at permeability, hydrophobic and aging resistance [12]. The SiO_2_/PVDF and non-woven (PP) PVDF composites are prepared by solution casting; the mechanical properties of the composites are studied by a uniaxial tensile test. The thermal shrinkage and combustion properties of the composites are analyzed by high temperature environment or combustion experiments. The microstructure of the composite is characterized by scanning electron microscopy (SEM), X-ray diffraction (XRD), energy dispersive spectrometer (EDS), thermogravimetric (TG), etc. The composite material prepared by this process has the advantages of higher strength and thermal stability.

## 2. Materials and Methods

### 2.1. Materials

The materials used in the experiment were PVDF, SiO_2_, N, N-dimethyl formamide (DMF, AR) and silane coupling agent (KH550, AR). PVDF was purchased from Shanghai O-Fluorine Chemical Technology Co. Ltd. (Shanghai, China). SiO_2_ with a 200 nm average diameter was purchased from Xilong Scientific Co. Ltd. (Shantou, China). and polypropylene non-woven fabric (PP, Shengye Co. Ltd., Jinhua, China, surface density 40 g/m^2^). Powder materials such as PVDF need to be dried in the oven before application. The chemicals were used without further purification in this work.

### 2.2. Preparation of Composite Films

The preparation process, pattern, form and size of composite materials are shown in Figure 1. The manufacturing equipment for composite samples was found in our previous work [17]. In this paper, PP/SiO_2_/PVDF composite was prepared by solution casting method. First, PVDF powder was dried in a drying oven, then added into the DMF solution, and dissolved by using a magnetic stirrer (MYP11-2, Chijiu Co., Shanghai, China). Second, a certain amount of silica powder was added to the liquid mixture and stirred at 60 °C for 3 h. Ultrasonic oscillator (ps-25t Jarcon Co., Ltd., Shenzhen, China) was used to disperse the silica powder for 10–15 min. In this experiment, two cycles of strong mixing and ultrasonic vibration were used to obtain the fully mixed solution for next step. Third, the polypropylene non-woven fabric was placed in the mold, and the flat surface was coated with the solution prepared in previous step. Finally, heat and dry the material in a drying oven. 

Based on our preliminary study, a perfect morphologic integrity and a smooth surface of PP/PVDF were maintained when the mass ratio between PP and PVDF was 10 wt %, leaving SiO_2_ as a variable with mass fraction 2 wt %, 4 wt %, 6 wt % and 8 wt %. The thermal–mechanical coupling experiment was carried out after cutting the sample properly.

### 2.3. Materials Characterization

The uniaxial tensile tests were carried out with Zwick Roell (Ulm, Germany) tensile tester. The thermal properties of the composites were analyzed by uniaxial tensile test under different temperatures and thermogravimetric analysis (TG, PerkinElmer, Waltham, MA, USA). The microstructure of the composite was observed by a scanning electron microscope (SEM, Oberkochen, Germany) and energy disperse spectroscopy (EDS, Oberkochen, Germany).

## 3. Results

### 3.1. Analysis of Mechanical Properties of Composite Materials

#### 3.1.1. Uniaxial Tensile Mechanical Properties of PP/SiO_2_/PVDF Composites

The lithium-ion battery will release heat in the process of application, meanwhile, the separator shows corresponding expansion and contraction. If the tensile strength of separator can not satisfy the requirement, it can break and resulting short-circuit of the battery. The key problem to improve the comprehensive properties of modified PP/SiO_2_/PVDF composite membrane is to combine the advantages of different materials. In this part, uniaxial tensile test was carried out using a Zwick Roell universal testing machine at room temperature according to the ASTM D 882-2012 standard. After preparation as mentioned before, the composites materials was cut to a sample with the size of 150 mm × 10 mm, and the thickness of the sample was measured with a screw micrometer. The length of the test area was 50 mm, as shown in Figure 1. Before the tensile test, the effective test area of the sample (length 50 mm) was marked, then, two ends of the sample was loaded by a tensile rate 5 mm/min. Five tests was carried out on each modified composite sample, and the stress and strain data of the test was recorded by the testing machine. 

As shown in Figure 2, the samples collected after the test are uneven. There were different forms of radians at the fracture surfaces, and some white filaments connecting the two fracture surfaces after the test could be observed. This is mainly due to the fact that PP non-woven fabric is composed of directional or random fibers, which are prepared by high temperature melting, spinning, hot pressing and other processes. The tensile ductility was relatively large, and the spinning could be observed by the microscope clearly. 

The uniaxial tensile stress–strain curve of PP/SiO_2_/PVDF composite is shown in Figure 3. From the whole curve, it can be seen that the curve conforms to the description of classical elastic-plastic curve. Table 1 shows the stress–strain values of pure PP non-woven, PP/PVDF and PP/SiO_2_/PVDF composites. The maximum stress and tensile strain of pure PP non-woven were 7.49 and 1.221 MPa, respectively. The maximum stress of PP/PVDF composite increased to 11.661 MPa, but the maximum strain decreased to about 0.428. In our previous research [17], it is found that the SiO_2_ filler can enhance the mechanical properties of PVDF polymer and improve the porosity. Similarly, the mechanical properties of PP/SiO_2_/PVDF modified by SiO_2_ nano-particles increased significantly. Especially when the content of SiO_2_ was 4 wt %, the maximum stress of PP/SiO_2_/PVDF was 18.314 MPa, which was 244.42% higher than that of pure PP. However, in case of a larger content of SiO_2_ (>4 wt %), the maximum stress and strain of the composite decreased with the increasing of content of SiO_2_.

#### 3.1.2. Effect of Temperature on the Mechanical Properties of PP/SiO_2_/PVDF Composites

In addition to the analysis of the mechanical properties of the composite at room temperature, we should also consider the thermal–mechanical coupling problem. Since the temperature of a lithium-ion battery will rise in the charging and discharging process, at the same time, the separator material is also in a stress state. It is very important to study the influence of temperature on the stress and strain of composite materials for their engineering application. In this study, uniaxial tensile tests were carried out at 50 and 80 °C considering the working condition of the lithium-ion battery. The stress–strain curve was obtained by calculating the experimental load and displacement data, as shown in Figure 4. By comparing the maximum stress of composite materials at different temperatures, it was found that the load bearing capacity of composite materials became worse for a higher test temperature. The composites at high temperature (80 °C) were harder and less elastic than those at room temperature, meanwhile, the maximum strain decreased. At 50 °C, the stress value of PP/SiO_2_/PVDF (4 wt %) composite decreased from 18.314 (at room temperature) to 12.303 MPa. With an increase of temperature to 80 °C, the stress was further reduced to 12.254 MPa. The comparison of the maximum stress-strain values of different materials at different temperatures is shown in Table 2. The results indicate that SiO_2_ particles could improve the load-bearing capacity of the composite, and make it possible to work in a relatively high temperature environment.

### 3.2. Thermal Stability Analysis

Thermal stability of the composite is very important for its safe and stable application in lithium-ion batteries [16]. Figure 5 shows the thermal shrinkage properties of pure PP non-woven, PP/SiO_2_/PVDF composites and SiO_2_/PVDF films at different temperatures. After heat treatment at 120 °C for 0.5 h, there was no obvious shrinkage of the four types of composites (PP, PP/PVDF, PP/SiO_2_/PVDF and SiO_2_/PVDF). When the heat treatment temperature reached 150 °C, pure PP non-woven fabric appeared to be edge buckling, the heat shrinkage was close to 20%, and the color did not change, as shown in Figure 5a. The other two materials had basically no change in size, but the color gradually became yellow. When the temperature increased to 170 °C, the heat shrinkage of pure PP non-woven increased to 50% of original sample obviously, which could not keep the original square shape. The volume of the PP/SiO_2_/PVDF (4 wt %) composite began to shrink, as shown in Figure 5c. When the temperature increased to 180 °C, pure PP non-woven fabric disappeared after complete thermal decomposition, the shrinkage of the PP/SiO_2_/PVDF (4 wt %) composite also reached about 50%, and the surface buckled and hardened. SiO_2_/PVDF composites bent at 180 °C. However, the volume did not shrink significantly. Some samples could not keep the square shape. This can be attributed to the high temperature resistance of SiO_2_ nano-materials, which can resist the thermal weight loss at a certain temperature. 

After the test of high temperature heating in the previous step, we further carried out the flame combustion test. A similar phenomenon was found in the combustion test [18]. The combustion properties of different composites are shown in Figure 6. When approaching the flame, PP non-woven material shrunk immediately and burnt completely within 3 s. The SiO_2_/PVDF material in Figure 6c had excellent flame retardant performance and could not be ignited at the end of the experiment. Especially when SiO_2_ was added, the combustion of the composite would be inhibited greatly. It was found that combustion of PP/SiO_2_/PVDF composites was greatly hindered by the presence of SiO_2_ and PVDF particles. The duration from close to the flame to the start of combustion was close to 26 s. This obviously helped to reduce the risk of combustion of the battery separator due to a high temperature.

TG analysis is one of the effective methods to study the mass loss, thermal stability and composition of materials in different thermal treatment condition. In this study, the heating rate of TG analysis was 10 °C/min, and the temperature was ranging from 30 to 800 °C. Figure 7 shows TG analysis of the PP/SiO_2_/PVDF composite filled with SiO_2_ (4 wt %). At the beginning of the test, the PP/SiO_2_/PVDF sample mass was 9.497 mg at room temperature, i.e., 30 °C. The carbon residual mass was 19.07% at 800 °C in nitrogen atmosphere. Taking SiO_2_/PVDF (4 wt %) as comparison, the residual mass rate (30.80%) was higher than that of PP/SiO_2_/PVDF. TG-DTG curves shows that the weight loss of the PP/SiO_2_/PVDF (4 wt %) composite membrane began at about 420 °C along with the first degradation of the material. The slope of the curve increased significantly when the temperature increased from 450 to 480 °C, reflecting the accelerated weight loss of the composite film. Compared with the TG weight loss curve of SiO_2_/PVDF (4 wt %) composite films, there was only one significant decline stage in the 440–450 °C range means thermal decomposition process was completed at one time. Then, the DTG curve was obtained by the first derivative of TG curve. It was observed that the peak of SiO_2_/PVDF (4 wt %) and PP/SiO_2_/PVDF (4 wt %) appeared at 463.94 and 460.31 °C, respectively. This difference was relative to the addition of PP.

### 3.3. Microstructure Characterization

#### 3.3.1. SEM and EDS Analysis 

Microscopic imaging, i.e., SEM, can be used to study the geometrical characteristics of microstructures of samples. In order to investigate the composition of the composite and observe the dispersion and mixing of SiO_2_ particles in PP/PVDF matrix, SEM observation was carried out on the sample. Figure 8a-b depicts the SEM image of the composite surface of pure PP non-woven fabric. It can be seen from the figure that pure polypropylene was randomly arranged to form a fiber network structure, and partially compressed into a slightly compact sheet. As a result, there were a lot of holes on the surface of pure PP non-woven fabric. SiO_2_ particles with a particle size between 150 and 250 nm were evenly distributed on the surface of the membrane after pouring and modifying SiO_2_/PVDF solution, as shown in Figure 8c. The surface was rough, meanwhile, some fibers had slight wrinkles. The results show that SiO_2_/PVDF composite was fixed on PP non-woven fabric perfectly.

Figure 9 is the SEM result of cross-section of composite after the tensile test. As shown in Figure 9a,b, it can be clearly observed that the broken non-woven filament fiber of PP/SiO_2_/PVDF (4 wt %) had small granular protrusions and folds on its cross-section. More important, the filaments became straight from the bending state when the material had a large strain. In contrast, Figure 9c,d shows the cross section of the SiO_2_/PVDF composite. Some uniform round pores with a different geometric form and size could be observed. Meanwhile, the comparison indicates that the porosity and connectivity of SiO_2_/PVDF was reduced. This phenomenon would lead to the change of porosity and permeability of the two materials. Generally, there were differences in the size, number and shape of the pores observed in the cross section of PP/SiO_2_/PVDF and SiO_2_/PVDF composites. Proper porosity is a prerequisite for lithium ion permeation or exchange in batteries.

The distribution of different elements in a certain area of the sample can be obtained by EDS, i.e., characterizing the two-dimensional distribution of X-ray intensity by using the scanning observation device. Figure 10 is an EDS image of the PP/SiO_2_/PVDF composite filled with 4 wt % SiO_2_. The analysis of the element composition indicates that (a) the composite material mainly contains the O, Si, C and F element and (b) the element distribution of each map in the observation area is relatively uniform. At the same time, the existence of SiO_2_ nanoparticles on the surface of the film could be confirmed by the Si mapping image. According to the test results in Figure 10b, the mass ratios of O, Si, C and F were 3.16%, 5.71%, 38.71% and 52.42%, respectively. C and F were the main elements of the composite resulting obvious wave crest and high content. Si was the main element of SiO_2_, and its content was also reasonable according to the experimental design.

#### 3.3.2. XRD Analysis

The properties of materials are strongly dependent on the arrangement of atoms. As for PVDF prepared by different experimental methods, five kinds of crystalline phases have been extensively observed and studied. XRD is the main method of the crystal phase analysis. XRD and related MDI jade V6.5 software analysis results of PVDF and PP/SiO_2_/PVDF are shown in Figure 11. Based on the XRD results and the basic type of elements in composites, appropriate data in the PDF card was selected and compared with the XRD results, as shown in Figure 11a. The peaks of 2θ = 14.187°, 17.054° were found in the material PDF cards corresponding to PP, and 2θ = 42.464° corresponded to SiO_2_. The diffraction peaks of PVDF were observed at 2θ = 18.469° and 26.586°, indicating the existence of an α crystals phase. Generally, the existence of β crystal was coincident with the wave peak at 2θ = 20.827 ° and 36.322°. Compared with the XRD curve of pure PVDF, the crystal phase of PVDF containing PP and SiO_2_ had not changed.

## 4. Conclusions

PP has the advantages of a low price, no pollution and high porosity. However, the degradation speed of PP non-woven fabric leads to a short service life, poor thermal stability and mechanical properties. It can not be directly used in an ion battery. PVDF has been used in liquid filtration due to its high filtration precision, long service life, strong anti pollution ability and stable product performance. In this paper, a PP/SiO_2_/PVDF composite was prepared by using PP non-woven fabric as the matrix, which had the following characteristics:

(1) PP/SiO_2_/PVDF had high tensile strength. The stress–strain relationship of the composite was obtained by a tensile test. When SiO_2_ content was 4 wt %, the maximum stress was 18.314 MPa, which was 244.42% higher than that of pure PP non-woven fabric (7.493 MPa). The maximum strain of the modified composite decreased to 0.294.

(2) It was found that temperature had a great influence on the mechanical properties of the composite through the thermal–mechanical coupling experiment. The stress and strain decreased with the increase of temperature. At room temperature, 50 and 80 °C, the stress of the PP/SiO_2_/PVDF (4 wt %) composite was 18.314, 12.303 and 12.254 MPa, respectively. The tensile properties of composites with 4 wt % content at different temperatures were still better than those of other samples.

(3) The SiO_2_/PVDF mixture could be well attached to the surface of non-woven fiber and form a package, with good dispersion and no agglomeration. After modification, the macropores of pure PP non-woven had changed into uniform small pores, which was beneficial to improve the quality of water filtration or infiltration capacity of lithium ion during battery charging and discharging. 

(4) The PP/SiO_2_/PVDF composite had good thermal shrinkage. It was found that the addition of SiO_2_ could improve the resistance to deformation under high temperature and stability of the composite. The TG-DTG curve also shows the effect of SiO_2_ on melting point and thermal weightlessness of the material.

## Figures and Tables

**Figure 1 polymers-12-00481-f001:**
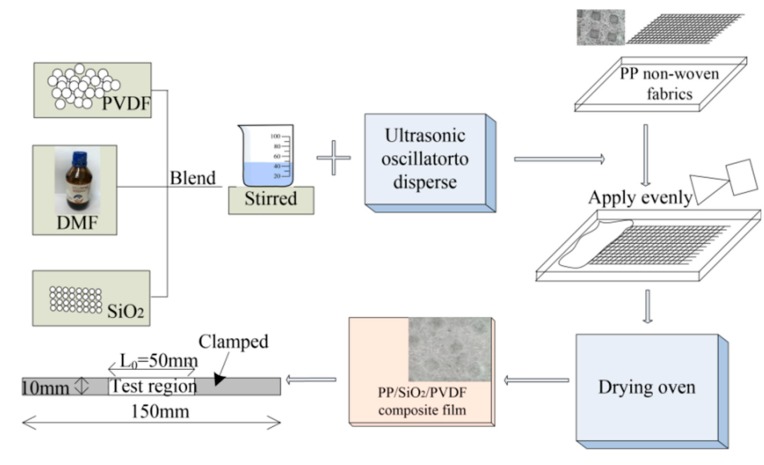
Composite material production flow chart.

**Figure 2 polymers-12-00481-f002:**
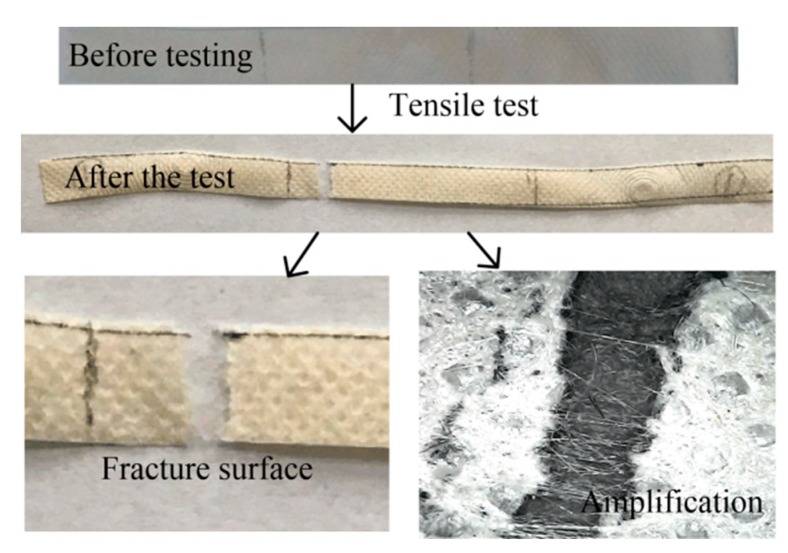
Samples before and after the tensile test.

**Figure 3 polymers-12-00481-f003:**
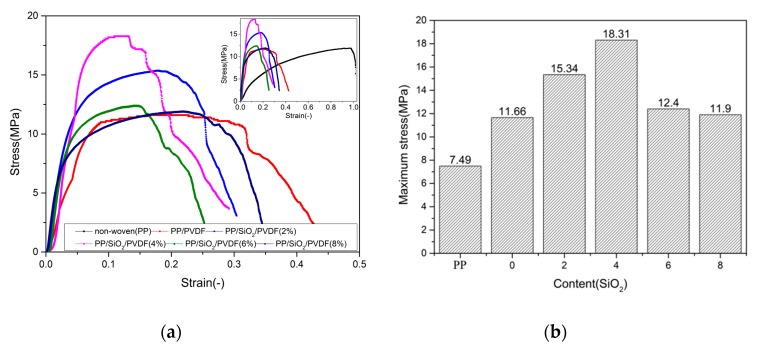
Stress strain curve and bar chart of tensile test: (**a**) stress strain curve and (**b**) maximum stress bar chart.

**Figure 4 polymers-12-00481-f004:**
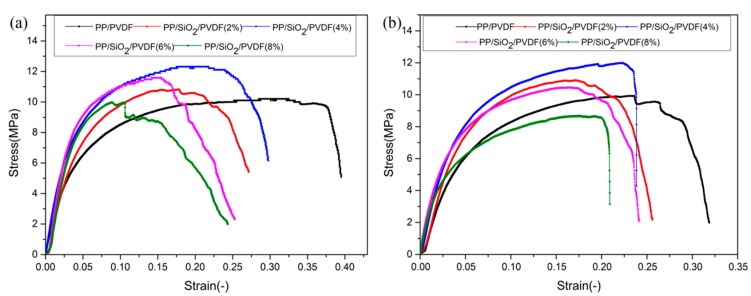
The stress–strain curves of composites at different temperatures: (**a**) 50 °C and (**b**) 80 °C.

**Figure 5 polymers-12-00481-f005:**
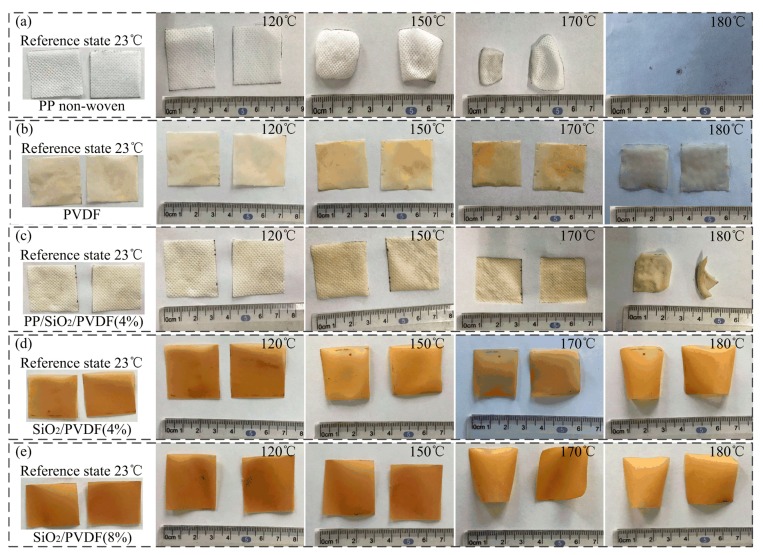
Thermal shrinkage properties of the film: (**a**) thermal shrinkage of PP non-woven materials under different heat treatment temperature; (**b**) PVDF; (**c**) PP/SiO_2_/PVDF (4%); (**d**) SiO_2_/PVDF (4%) and (**e**) SiO_2_/PVDF (8%).

**Figure 6 polymers-12-00481-f006:**
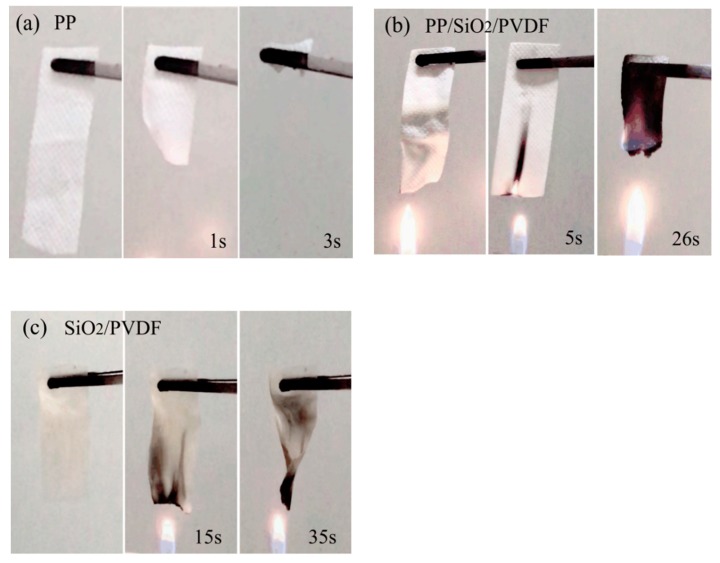
Combustion behavior of the materials: (**a**) PP; (**b**) PP/SiO_2_/PVDF and (**c**) SiO_2_/PVDF.

**Figure 7 polymers-12-00481-f007:**
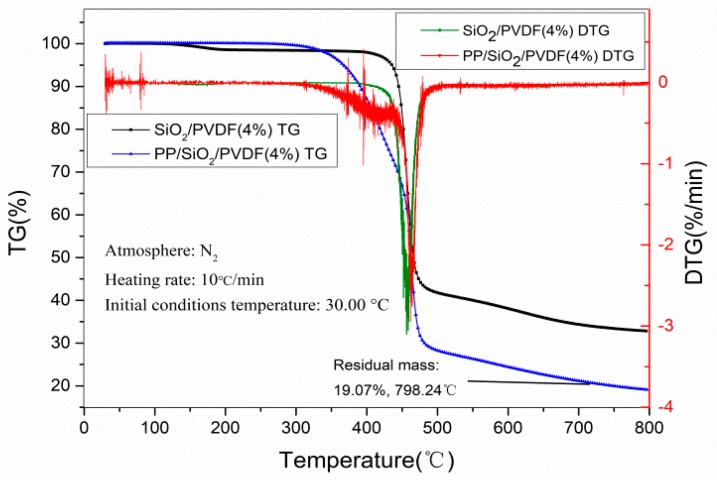
Thermal analysis curve of the PP/SiO_2_/PVDF and SiO_2_/PVDF composite.

**Figure 8 polymers-12-00481-f008:**
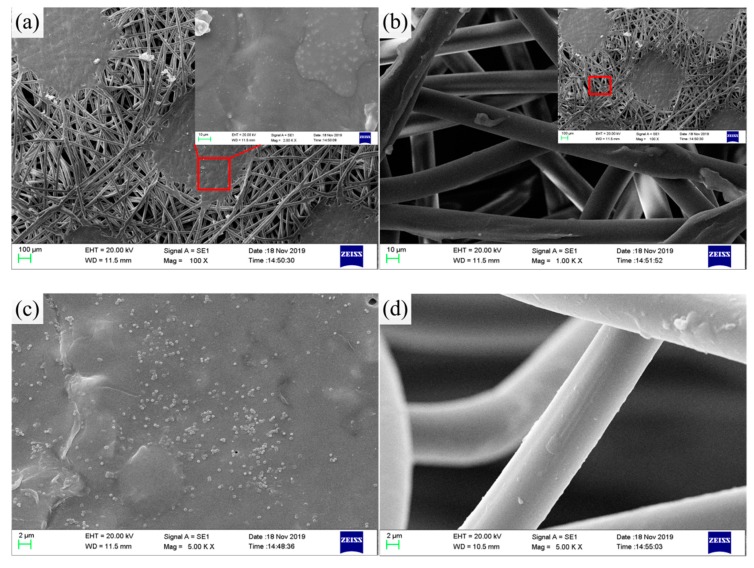
SEM of the composite surface: (**a**) PP non-woven at 100×; (**b**) PP non-woven at 1000×; (**c**) PP/SiO_2_/PVDF composite with 4 wt % SiO_2_ and (**d**) PP/SiO_2_/PVDF (4 wt %) magnified by 5000×.

**Figure 9 polymers-12-00481-f009:**
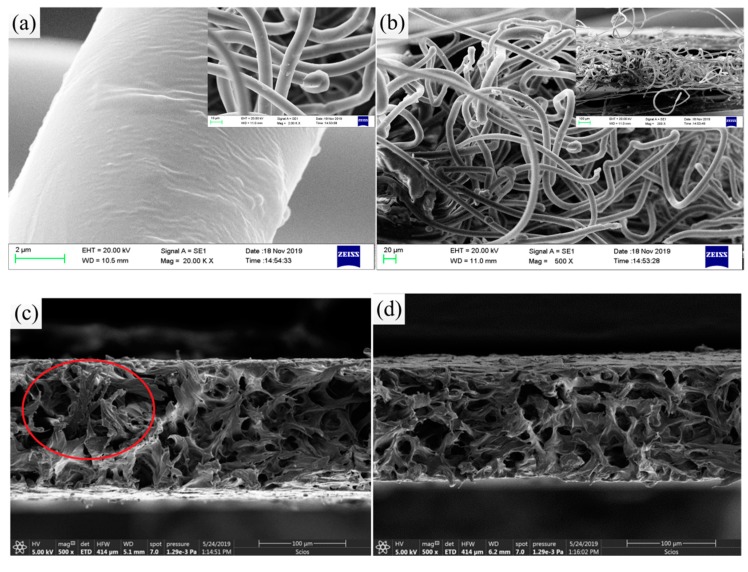
SEM of the cross section: (**a**) PP/SiO_2_/PVDF (4 wt %) magnified by 20,000×; (**b**) PP/SiO_2_/PVDF (4 wt %) magnified by 500×; (**c**) SiO_2_/PVDF (6 wt %) magnified by 500× and (**d**) SiO_2_/PVDF (4 wt %) magnified by 500×.

**Figure 10 polymers-12-00481-f010:**
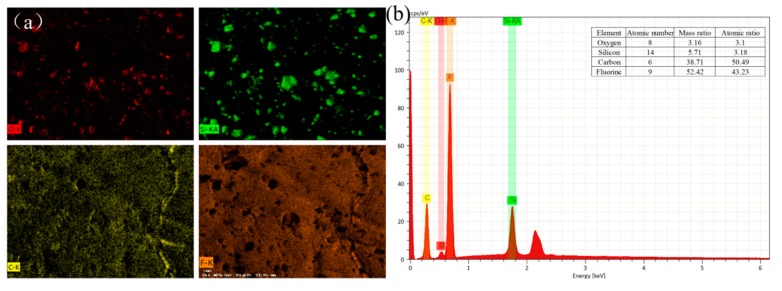
Energy dispersive spectrometer (EDS) analysis diagram of the PP/SiO_2_/PVDF composite with 4 wt % SiO_2_: (**a**) EDS mapping of O, C, Si and F and (**b**) elemental EDS analysis.

**Figure 11 polymers-12-00481-f011:**
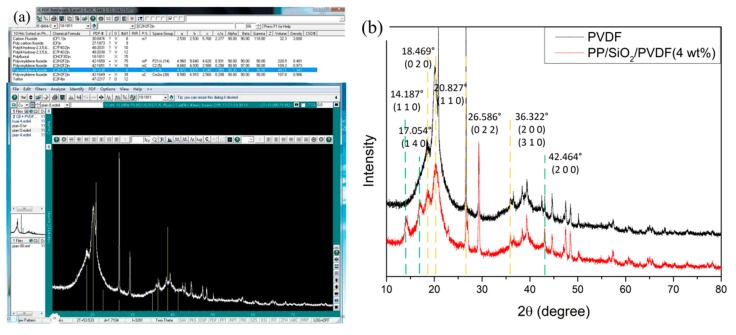
XRD analysis of samples: (**a**) Jade (XRD analysis software) and (**b**) curve of XRD.

**Table 1 polymers-12-00481-t001:** Comparison of maximum stress and strain.

Materials	Strain (-)	Stress (MPa)	Contrast of Stress (%)
PP	1.221	7.493	-
PP/PVDF	0.428	11.661	155.62
PP/SiO_2_/PVDF (2%)	0.389	15.346	204.80
PP/SiO_2_/PVDF (4%)	0.294	18.314	244.42
PP/SiO_2_/PVDF (6%)	0.252	12.396	165.43
PP/SiO_2_/PVDF (8%)	0.344	11.901	158.82

**Table 2 polymers-12-00481-t002:** Maximum stress and strain.

Materials	Stress at 50 °C (MPa)	Strain at 50 °C (-)	Stress at 80 °C (MPa)	Strain at 80 °C (-)
PP/PVDF	10.183	0.395	9.394	0.319
PP/SiO_2_/PVDF (2%)	10.826	0.281	10.906	0.256
PP/SiO_2_/PVDF (4%)	12.303	0.298	12.254	0.239
PP/SiO_2_/PVDF (6%)	11.951	0.253	10.764	0.241
PP/SiO_2_/PVDF (8%)	10.102	0.252	9.070	0.240

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
