# Peer review of "Analysis of Thermal–Mechanical Properties of Silicon Dioxide/Polyvinylidene Fluoride Reinforced Non-Woven Fabric (Polypropylene) Composites"

_polymers, 2020, doi:10.3390/polym12020481_

Round 1

Reviewer 1 Report

The manuscript “Analysis of thermal-mechanical properties of silicon dioxide/polyvinylidene fluoride reinforced non-woven fabric (polypropylene) composite” investigate mechanical property and thermal property of SiO2/PVDF composite prepared by solution casting method. I feel this manuscript is not ready for publish any other journal. Authors should put more and careful effort into preparing their manuscript. My arguments are shortly listed below.

In “2.2 preparation of composite films” section, instruction of manuscript preparation are still remain. Many unnecessary information are included in the manuscript. For example, “2.3 materials characterization” section, explain of each measurement systems are not required. The authors should perform more systematic and statistics experiment. In figure 5, there is no information of each samples. In figure 7, there is no mention about what the author wanted to claim. In addition, there is no results from the comparing samples. In figure 8 and 9, 10, 11 SEM images, EDS analysis, and XRD patterns do not deliver any messages. Further English polishing and typo corrections are required before publication

Reviewer 2 Report

This study introduced a solution casting method to prepare the composite film with PP non-woven fabric as the support and silicon dioxide/polyvinylidene fluoride (SiO2/PVDF) as the filler. However, some comments should be addressed before accepting:

L122 to L133, please remove the irrelevant sentences “Materials and Methods should be described with” etc. Please check carefully before submission. In Figure 2, there’s 5 samples in Fig. 2(a) while only 4 in Fig. 2(b), why is one missing? What’s are these samples? Any differences among samples numbering 1-3? How long did it take for them to break? This information should be illustrated in your figures and text. Why temperature of 50 deg and 80 deg were selected in part 3.1.2, reason should be given. The names of samples should be labelled in Figure 5. What’s the percentage of SiO2 used in PP/SiO2/PVDF in tests in part 3.2? From your point of view, is there an optimal percentage of SiO2 in PP/SiO2/PVDF? If so, please discuss and investigate more in your paper.

Round 2

Reviewer 1 Report

The manuscript “Analysis of thermal-mechanical properties of silicon dioxide/polyvinylidene fluoride reinforced non-woven fabric (polypropylene) composite” investigate mechanical property and thermal property of SiO2/PVDF composite prepared by solution casting method. This manuscript is well written, and it has sufficient novelty after revision. Therefore, I would recommend this manuscript for the possible publication in Polymer

Reviewer 2 Report

The paper can be accepted.